# Co-factor-free aggregation of tau into seeding-competent RNA-sequestering amyloid fibrils

Pijush Chakraborty [1], Gwladys Rivière[1], Shu Liu[2], Alain Ibáñez de Opakua[1], Rıza Dervişoğlu [3],
Alina Hebestreit[2], Loren B. Andreas [3], Ina M. Vorberg [2,4] & Markus Zweckstetter [1,3 ✉]

Pathological aggregation of the protein tau into insoluble aggregates is a hallmark of neurodegenerative diseases. The emergence of disease-specific tau aggregate structures termed tau strains, however, remains elusive. Here we show that full-length tau protein can be aggregated in the absence of co-factors into seeding-competent amyloid fibrils that sequester RNA. Using a combination of solid-state NMR spectroscopy and biochemical experiments we demonstrate that the co-factor-free amyloid fibrils of tau have a rigid core that is similar in size and location to the rigid core of tau fibrils purified from the brain of patients with corticobasal degeneration. In addition, we demonstrate that the N-terminal 30 residues of tau are immobilized during fibril formation, in agreement with the presence of an N-terminal epitope that is specifically detected by antibodies in pathological tau. Experiments in vitro and in biosensor cells further established that co-factor-free tau fibrils efficiently seed tau aggregation, while binding studies with different RNAs show that the co-factor-free tau fibrils strongly sequester RNA. Taken together the study provides a critical advance to reveal the molecular factors that guide aggregation towards disease-specific tau strains.

[1] German Center for Neurodegenerative Diseases (DZNE), Göttingen, Germany. [2] German Center for Neurodegenerative Diseases (DZNE), Bonn, Germany.
[3] Department for NMR-based Structural Biology, Max Planck Institute for Biophysical Chemistry, Göttingen, Germany. [4] Rheinische Friedrich-Wilhelms-Universität, Bonn, Germany. ✉email: Markus.Zweckstetter@dzne.de

Pathological aggregation of the microtubule-binding protein tau (Fig. 1a) into amyloid fibrils is a hallmark of different neurodegenerative diseases collectively termed tauopathies[1]. To date, tau amyloid structures associated with four different tauopathies (Alzheimer's disease, Chronic traumatic encephalopathy, Corticobasal degeneration (CBD; Fig. 1b) and Pick's disease) have been determined[2–6]. Strikingly, structures are homogeneous within one disease, but significantly different between different diseases, suggesting a critical interplay between the amyloid structure, the nature of the disease and its propagation. The distinct aggregate structures, termed amyloid "strains", can induce aggregation of tau in cellula and recapitulate pathological phenotypes when injected into animals[7]. So far, the emergence of such tau amyloid strains remains elusive, and the factors that drive tau aggregation towards a well-defined strain are unknown.

Recombinant tau monomers can efficiently form fibrils in vitro but only in the presence of negatively charged co-factors such as heparin[8,9]. The biological relevance of co-factor-induced fibrillization as an in vitro model of tau aggregation has, however, been questioned[10]. Indeed, cryo-electron microscopy (cryoEM) of heparin-induced fibrils of the longest isoform of tau (2N4R tau; Fig. 1a) demonstrated that the heparin-induced fibrils differ structurally from the tau filaments extracted from human patient brain (Fig. 1b, c)[10]. Another major drawback of the heparin-based in vitro fibrilization assay is the high negative charge of heparin: heparin-induced tau fibrillization has been extensively used to search for small molecules as tau aggregation inhibitors[11,12], potentially generating false hits due to electrostatic interactions between the small molecules and heparin.

Here we describe an approach to convert full-length tau protein into amyloid fibrils in the absence of heparin or other negatively charged co-factors. Using a combination of biochemical experiments and NMR spectroscopy we then provide evidence that the co-factor-free tau fibrils have structural properties that largely differ from those of heparin-induced tau fibrils. In addition, we show that the tau fibrils aggregated in the absence of heparin display certain properties of amyloid fibrils from patient material, including a similar size and location of the fibrillar core,

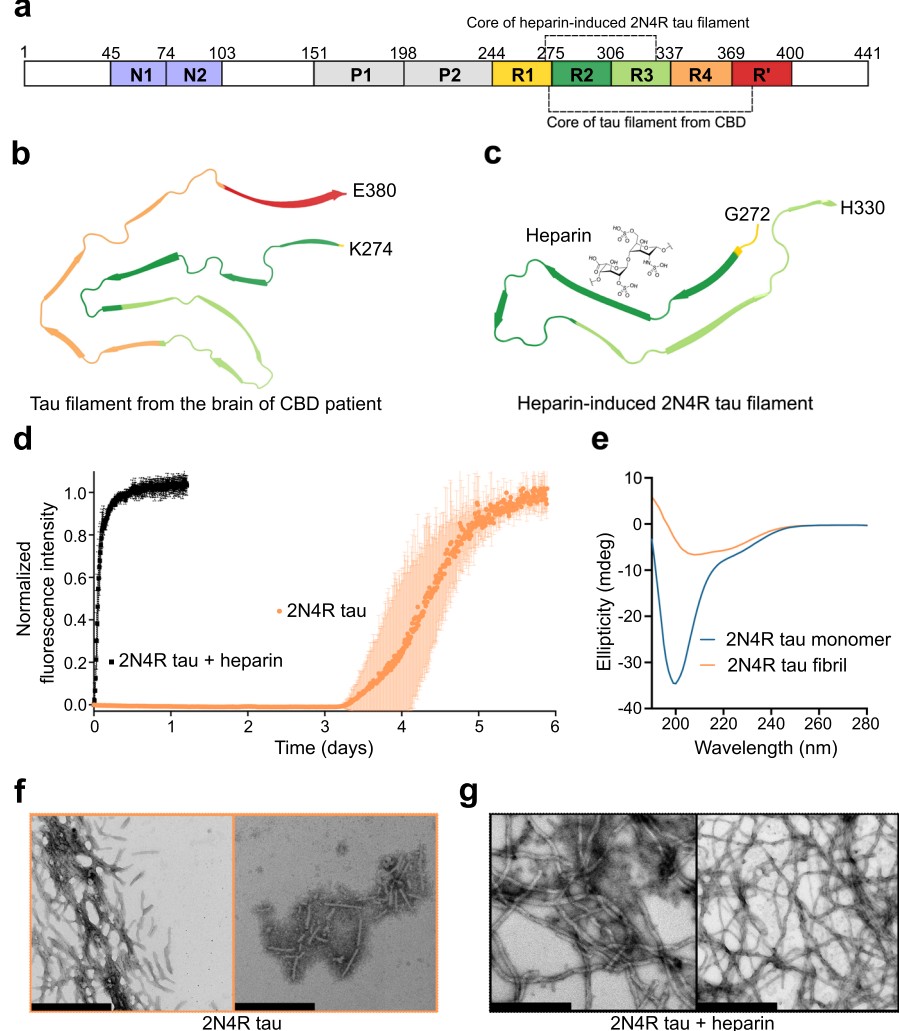

**Fig. 1 Amyloid fibrils of tau without co-factors. a** Schematic representation of the domain organization of 2N4R tau. N1 and N2 are two inserts subject to alternative splicing, P1 and P2 mark the proline-rich regions, and R1-R' are pseudo-repeats that bind to microtubules. Tau fibril cores from CBD-brain and heparin-induced 2N4R fibrils are marked by dashed lines (see also (**b**, **c**)). **b** CryoEM structure of tau filament (type 1) extracted from the brain of a patient with CBD (PDB code: 6TJO). **c** CryoEM structure of heparin-induced 2N4R tau fibrils (snake form; PDB code: 6QJH). A molecule of heparin is displayed to illustrate that these fibrils are formed in the presence of heparin. **d** Aggregation kinetics of 25 μM 2N4R tau with (black) and without (orange) heparin. Data are presented as mean ± standard deviation of $n = 3$ independent samples. **e** Circular dichroism spectra of 2N4R tau monomer and heparin-free fibrils. **f**, **g** Negative-stain electron micrographs of 2N4R fibrils aggregated without heparin (**f**) or with heparin (**g**). Scale bar, 500 nm. Similar micrographs have been observed for both 2N4R fibrils aggregated in presence/absence of heparin with ten independently aggregated samples.

a change in the molecular properties of the N-terminal antibody binding epitope and the strong and specific sequestration of RNA.

## Results

**Full-length tau fibrillizes without co-factors.** To achieve tau fibrillization in the absence of co-factors, we incubated 25 μM 2N4R tau at 37 °C in 25 mM HEPES, 10 mM KCl, 5 mM MgCl$_2$, 3 mM TCEP, 0.01% NaN$_3$, pH 7.2, buffer with polytetrafluoroethylene beads. To trigger aggregation, double orbital shaking at an interval of every 10 min was applied. Thioflavin-T was used to monitor the aggregation kinetics. 2N4R tau started aggregating after 3 days in the absence of heparin as compared to very rapid aggregation in presence of heparin (Fig. 1d). After about 6 days the ThT fluorescence saturated in the absence of heparin, indicating that heparin-free fibrillization of 2N4R was complete (Fig. 1d). The time-dependent ThT fluorescence of heparin-free tau aggregation displayed the typical sigmoidal kinetics of nucleation-dependent protein aggregation (Fig. 1d). By measuring the concentration of the monomeric protein left after reaching ThT saturation, we found that ~80% of 2N4R tau was aggregated (Supplementary Fig. 1).

To gain first insights into the structure of the aggregated 2N4R tau, we recorded circular dichroism (CD) spectra and negative-stain electron microscopy (EM) images. According to CD, monomeric tau prior to aggregation displays a spectrum (Fig. 1e), which is characteristic for intrinsically disordered proteins such as tau[13,14]. In contrast, the CD spectrum of the heparin-free 2N4R tau fibrils obtained after 6 days of aggregation (Fig. 1e) is typical for amyloid fibrils comprised of a ß-structure-rich core and flexible tails. Quantification of the CD spectrum estimated the ß-structure content as ~39%.

In the case of heparin-induced tau aggregation, it is known that this results in long tau fibrils that form an almost net-like arrangement on EM grids (Fig. 1g)[15,16]. In contrast, the 2N4R tau fibrils formed in the absence of heparin (but otherwise using the identical protocol, i.e., the same buffer, the same polytetrafluoroethylene beads and the same shaking procedure) are short and do not display an extended network on the EM grid (Fig. 1f). The overall morphological properties of tau fibrils obtained without and with heparin are thus very different (Fig. 1f, g).

**Protease-resistant core of heparin-free tau fibrils.** To obtain direct information about the fibrillar core, 2N4R fibrils were digested by trypsin to remove the fuzzy coat (Fig. 2a), followed by pelleting down the protease-resistant material through ultracentrifugation. SDS-PAGE gel analysis indicated a much longer core for 2N4R tau fibrils as compared to heparin-induced fibrils (Fig. 2b). The short trypsin-resistant core of heparin-induced fibrils is in agreement with previous data[16].

To further analyze the rigid core of the fibrils, we determined the sequence of the tau bands observed in SDS-PAGE by mass spectrometry. For heparin-induced fibrils, we detected a large number of peptides from residues ~260 to ~340 (Fig. 2c; black; Supplementary Fig. 2), consistent with structural analysis by cryoEM (Fig. 1c)[10]. For 2N4R fibrils, we detected a large number of peptides from residue ~280 to ~400 (Fig. 2c, orange; Supplementary Fig. 3). In addition, a few peptides for the C-terminal tail were detected, most probably because trypsin cuts the protein carboxy-terminally of K/R residues and there are only a few target residues (K395, R406, K438) for trypsin in this region.

To get more precise information about the protease-resistant core of the heparin-free 2N4R fibrils, especially to get more resolution at the C-terminus, we repeated the experiments with pronase. Pronase is a mixture of several endo- and exoproteases that can digest a protein into individual amino acids. 2N4R fibrils were digested by pronase and the resistant fibril core was pelleted down by ultracentrifugation. After excising the pronase-resistant band with trypsin, we detected peptides up to residue 379 (Supplementary Fig. 4).

**Rigidification of the N-terminal epitope of pathological tau.** Next, we aggregated uniformly $^{13}$C/$^{15}$N-labeled 2N4R tau both in the absence and presence of heparin and recorded $^1$H-$^{15}$N Insensitive nuclei enhanced by polarization transfer (INEPT) spectra under conditions of magic angle spinning (Fig. 2d and Supplementary Fig. 5). INEPT experiments use scalar coupling for polarization transfer, thereby detecting only the highly dynamic residues in solid samples, i.e., the fuzzy coat in the case of tau fibrils (Fig. 2a, d). Residue-specific analysis of the INEPT spectra revealed a complete loss of signal from residues ~270 to ~400 in the case of 2N4R fibrils (Fig. 2e), in agreement with mass spectrometry analysis (Fig. 2c). In the case of heparin-induced fibrils, no INEPT signals were detected for residues ~260 to ~330. Because cryoEM resolves only the very rigid residues in cross-ß-structure at high resolution and has reported additional unidentified electron density in tau fibrils, the analysis suggests that in vitro-generated 2N4R fibrils, but not heparin-induced fibrils, have a similar core length as tau fibrils purified from CBD patient brain (residues ~274 to ~380 were resolved at high resolution; PDB code: 6TJO; marked in light blue in Fig. 2e).

The NMR analysis, however, provided further insights into the key regions of tau aggregation. Not only residues ~270 to ~400 were broadened beyond detection in the case of the heparin-free 2N4R tau fibrils, but also the amino-terminal 30 residues (Fig. 2e; marked in pink). This is most likely because of a dynamic interaction of the N-terminus of tau with the cross-ß-structure core[17]. The immobilization of the N-terminal 30 residues provides a structural basis for the specificity of antibodies that specifically detect pathological tau[18] and require two discontinuous epitopes located in the repeat region (residues 313–322) and at the N-terminus (residues 1–18)[18].

**Solid-state NMR of the fibrillar core.** Next, we probed the rigid core of the 2N4R tau fibrils. To this end, $^{13}$C-$^{13}$C correlations were recorded using cross-polarization solid-state NMR experiments (Fig. 3a). In agreement with mass spectrometry and the INEPT-based NMR analysis (Fig. 2), the $^{13}$C-$^{13}$C spectrum of 2N4R fibrils contained many more signals when compared to the spectrum of heparin-induced fibrils (Fig. 3a). This was particularly evident from the C$_\beta$ region of serine and threonine residues (Fig. 3a), indicating that more serine and threonine residues are located in the core of 2N4R fibrils (Fig. 3b, c). In addition, only one broad cross-peak was observed in the threonine C$_\gamma$ region of the heparin-induced fibrils, while about six cross peaks were present in the case of the 2N4R fibrils (Fig. 3a, selected zoom). Because only rigid residues are detected in the cross-polarization NMR experiments, the number of observed threonine cross peaks is consistent with the fibrillar core of 2N4R fibrils derived from mass spectrometry and INEPT-based NMR experiments (Fig. 3b, c).

To further characterize the 2N4R fibrils, we recorded 2D NCA as well as 3D NCACX and 3D NCOCX experiments. Due to strong signal overlap, however, only a few residues in the fibril core could be identified (Fig. 3d–f). This included the residue stretch from G303 to V306 (Fig. 3d–f) that is part of the cross-ß-structure core of tau fibrils purified from a CBD patient (Fig. 1b). Isolated peaks in the 2D RFDR and 2D NCA spectra of the heparin-free 2N4R tau fibrils show $^{13}$C and $^{15}$N line widths of

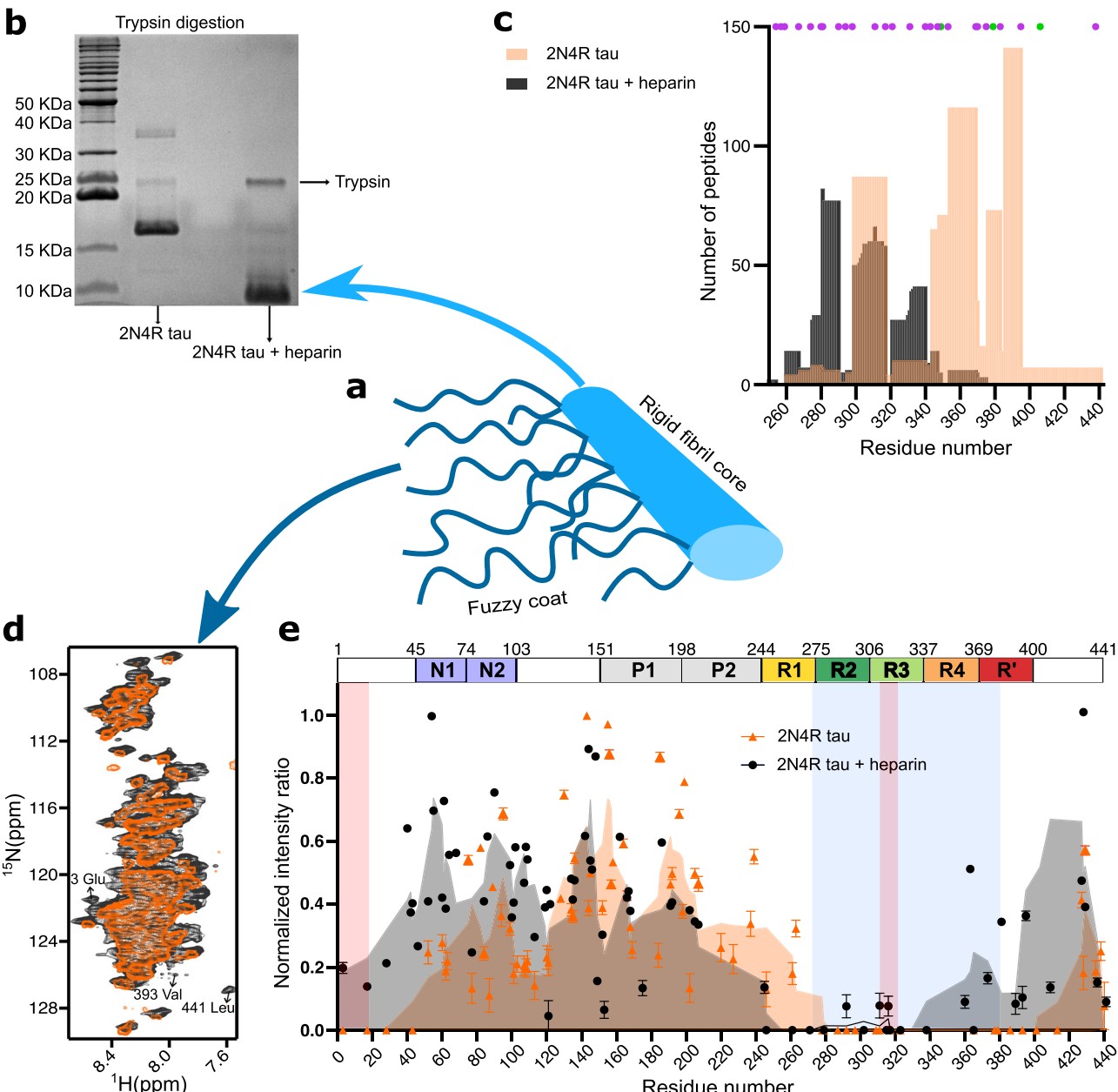

**Fig. 2 Heparin-free tau fibrils have an extended core and an immobile N-terminal antibody-binding epitope. a** Cartoon representation of the rigid core and dynamic (termed fuzzy) coat of tau fibrils. **b** SDS-PAGE gel of trypsin-digested tau fibrils formed in the absence or presence of heparin. The trypsin band is indicated. The result was reproducible for three independently performed experiments. **c** Number of peptides detected from the enzymatic digestion of the tau bands observed in SDS-PAGE in (**a**). The position of lysine and arginine residues of 2N4R tau are marked with purple and green dots, respectively. **d** Superposition of $^{1}$H-$^{15}$N INEPT spectra of 2N4R tau fibrils aggregated in the absence (orange) and presence (black) of heparin. **e** Intensity ratio plot of 2N4R tau fibrils aggregated in the absence (orange) and presence (black) of heparin. The intensity ratio was calculated by dividing the signal intensity of each residue in the fibril state by the monomeric state. The error of the intensity ratio for each residue was calculated from the signal-to-noise ratio of the cross peaks in the spectra. The rigid cross-ß-sheet core of the tau fibril extracted from a CBD patient brain (PDB code: 6TJO) is marked in light blue. The two discontinuous epitopes (residues 1–18 and residues 313–322) of antibodies that specifically detect pathological tau[18] are marked by red boxes.

0.6–0.8 ppm and 0.9–1.1 ppm, respectively, indicating structural homogeneity of the rigid core.

To gain further insight into the structural properties of 2N4R tau fibrils, we aggregated selectively labeled ($^{13}$C$_\gamma$ valine, $^{13}$C-ring phenylalanine, $^{15}$N histidine) 2N4R tau in the absence of heparin. 2D hCH and 2D hNH cross-polarization spectra demonstrated that the protein is $^{13}$C labeled only at the C$_\gamma$ of valine and the ring carbons of phenylalanine, and $^{15}$N-labeled at the backbone and side chain of histidine residues (Fig. 4a; Supplementary Fig. 6).

We then recorded Dynamic Nuclear Polarization (DNP)-enhanced solid-state NMR spectra in order to achieve maximum sensitivity. We used TEMTriPol-1[19,20] and observed a 12-fold DNP enhancement of the signal (Fig. 4b).

Next, DNP-enhanced 2D hCHHC spectra with 200 μs and 400 μs mixing were measured. The chosen mixing times will results in cross peaks between two $^{13}$C atoms that are about ≤4 Å and ≤6 Å apart, respectively[21]. In the spectra displayed in Fig. 4c, two cross peaks between the C$_\gamma$ of valine and the ring carbons of

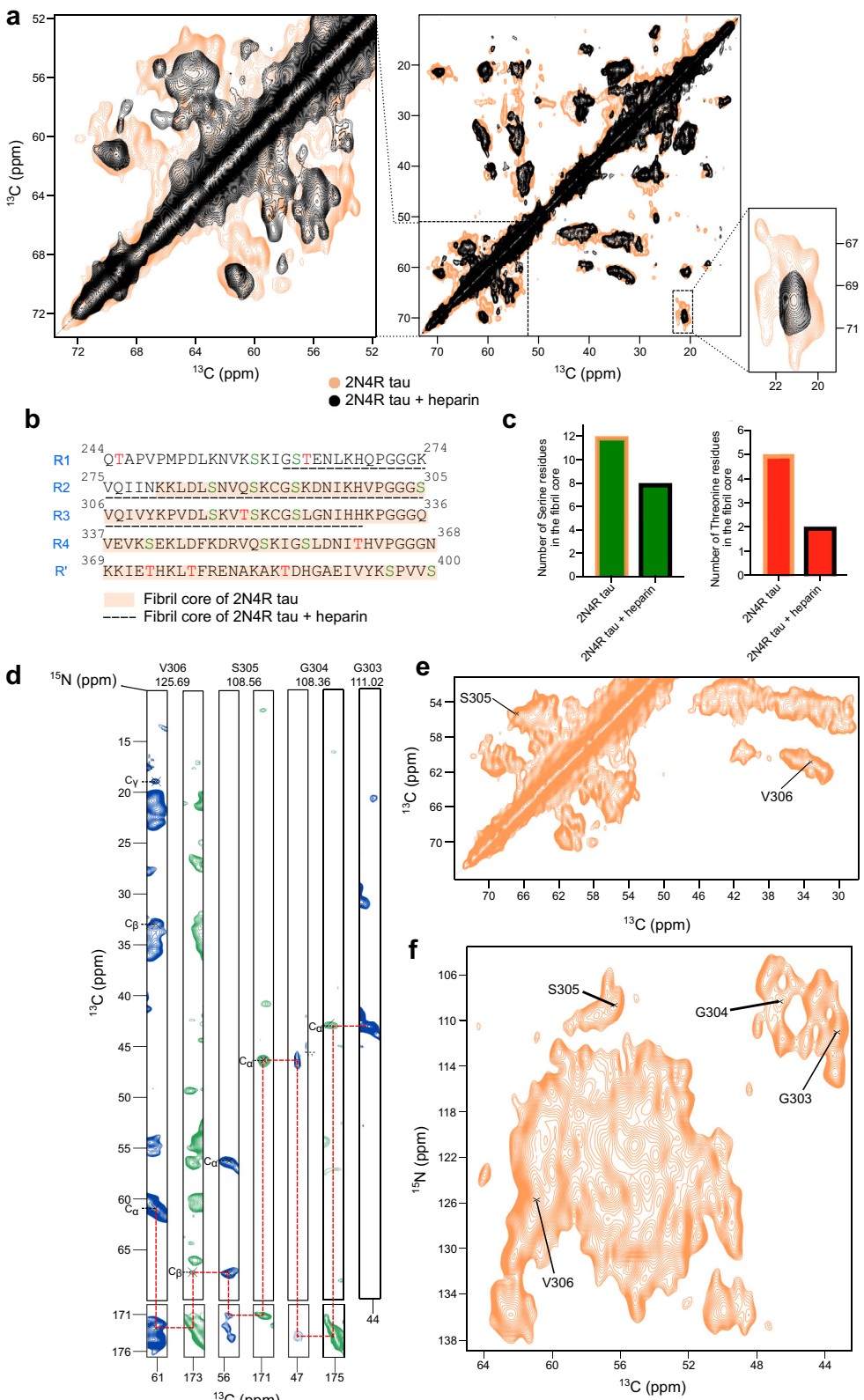

phenylalanine are present. The cross peaks are detected both below and above the diagonal (Fig. 4c), in agreement with the expected symmetric magnetization transfer. The presence of the two cross peaks suggests that the side chains of one or two valine residues are in close spatial proximity to the side chain of a phenylalanine residue. We also recorded DNP-enhanced 2D hNHHC spectra with 200 μs mixing and observed a cross peak between the $^{15}N$-ring of

histidine and $^{13}C_\gamma$ of valine (Fig. 4d), indicating that the aromatic ring of a histidine is in proximity (~4 Å) to the side chain of a valine residue in the structure of 2N4R fibrils.

The 2N4R tau sequence contains three phenylalanine residues, but only two (F346 and F378) are located in the rigid/semi-rigid core of 2N4R tau fibrils, which comprises residues ~270 to ~380 (Fig. 2). In the cryoEM structure of ex vivo CBD fibrils, F346

**Fig. 3 Solid-state NMR of the core of heparin-free tau fibrils. a** $^{13}$C-$^{13}$C RFDR spectra of 2N4R tau fibrils (middle panel), which were aggregated without (orange) or with heparin (black). The $C_\beta$ region of serine and threonine residues is highlighted on the left, the $C_\gamma$ region of threonine to the right. **b** Amino acid sequence of the repeat region of 2N4R tau. Serine residues are colored in green, threonine residues in red. The core of heparin-free 2N4R tau fibrils is highlighted in orange. The fibrillar core of heparin-induced 2N4R tau fibrils is underlined with black dashed lines. **c** Number of serine and threonine residues in the core of 2N4R tau fibrils that were formed in the absence or presence of heparin (according to (**b**)). **d**, Strips from 3D NCACX (blue) and 3D NCOCX (green) spectra of heparin-free 2N4R tau fibrils, illustrating the sequential assignment of residues. **e** Selected region of the $^{13}$C-$^{13}$C RFDR spectrum of heparin-free 2N4R tau fibrils, indicating the assignment of the cross peaks of residues S305 and V306. **f** 2D NCA spectrum of heparin-free 2N4R tau fibrils, indicating the assignment of the cross peaks of residues 303-306 (based on (**d**)).

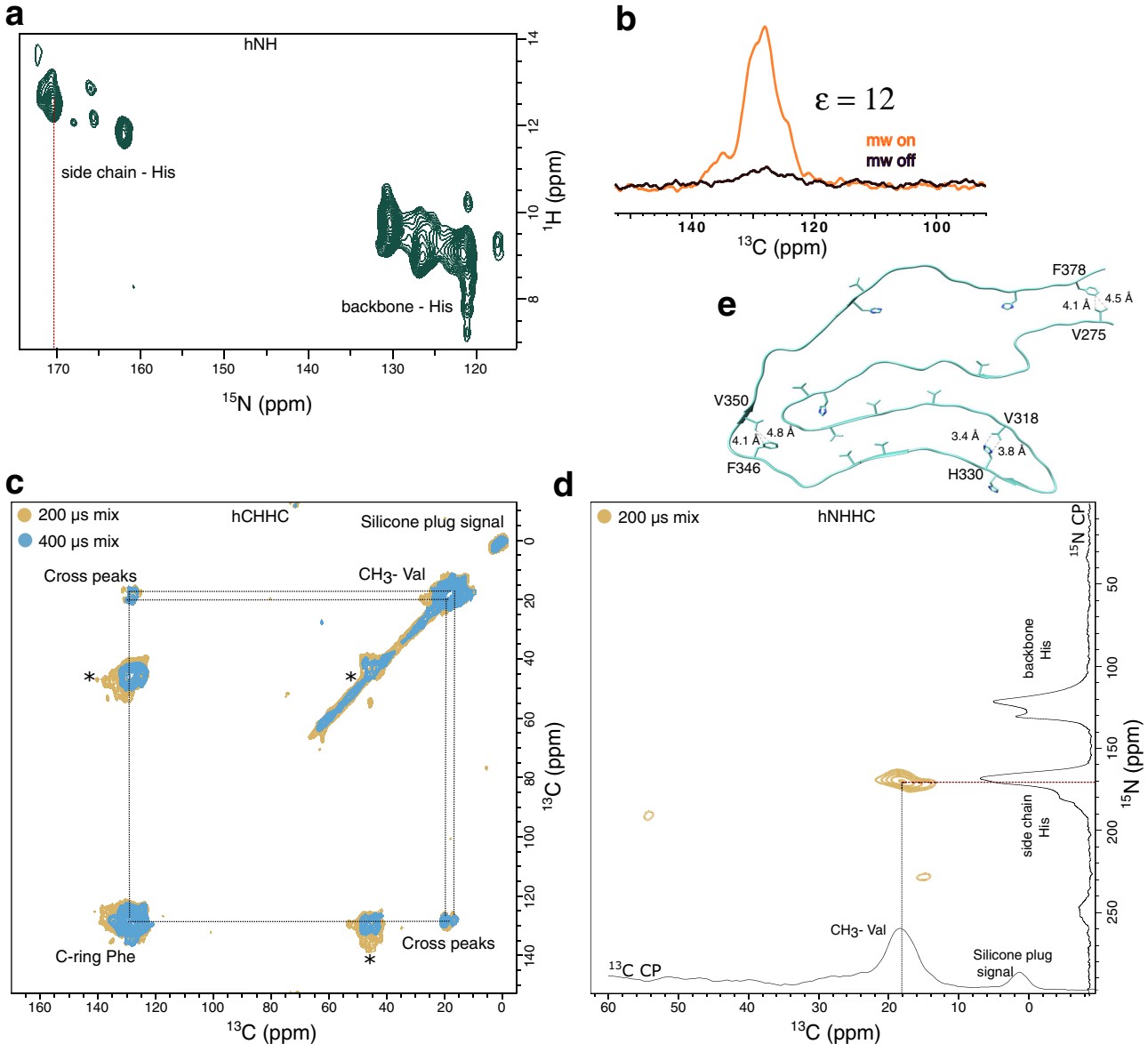

**Fig. 4 Residue type-specific contacts within the core of heparin-free 2N4R tau fibrils. a** Proton-detected $^{1}$H-$^{15}$N correlation solid-state NMR spectrum of heparin-free fibrils of 2N4R tau selectively labeled with $^{15}$N Histidine (and at the $^{13}C_\gamma$ of valine and the $^{13}$C-ring of phenylalanine). The spectrum was recorded at room temperature at a NMR spectrometer with 850 MHz $^{1}$H frequency using 55 kHz magic angle spinning (MAS) frequency. **b** Comparison of the $^{13}$C NMR signal of phenylalanine (ring carbons) when the microwave is turned on (orange) or off (black). The signal enhancement (ε) is displayed. Spectra shown in (**b**–**d**) were recorded at a temperature of 100 K at a NMR spectrometer with 600 MHz $^{1}$H frequency using 12.5 kHz MAS spinning. **c** Superposition of DNP-enhanced 2D hCHHC spectra of selectively labeled ($^{13}C_\gamma$ valine, $^{13}$C-ring phenylalanine, $^{15}$N histidine) heparin-free 2N4R tau fibrils with 200 μs (yellow) and 400 μs (sky blue) mixing time. Spinning side bands are labeled with *. **d** DNP-enhanced 2D hNHHC spectrum of selectively labeled ($^{13}C_\gamma$ valine, $^{13}$C-ring phenylalanine, $^{15}$N histidine) heparin-free 2N4R tau fibrils. **e** Location of valine, phenylalanine and histidine residues within the core structure of the tau filament (type 1) extracted from the brain of a patient with CBD (PDB code: 6TJO). Short-range distances between valine and phenylalanine, and histidine and valine are marked.

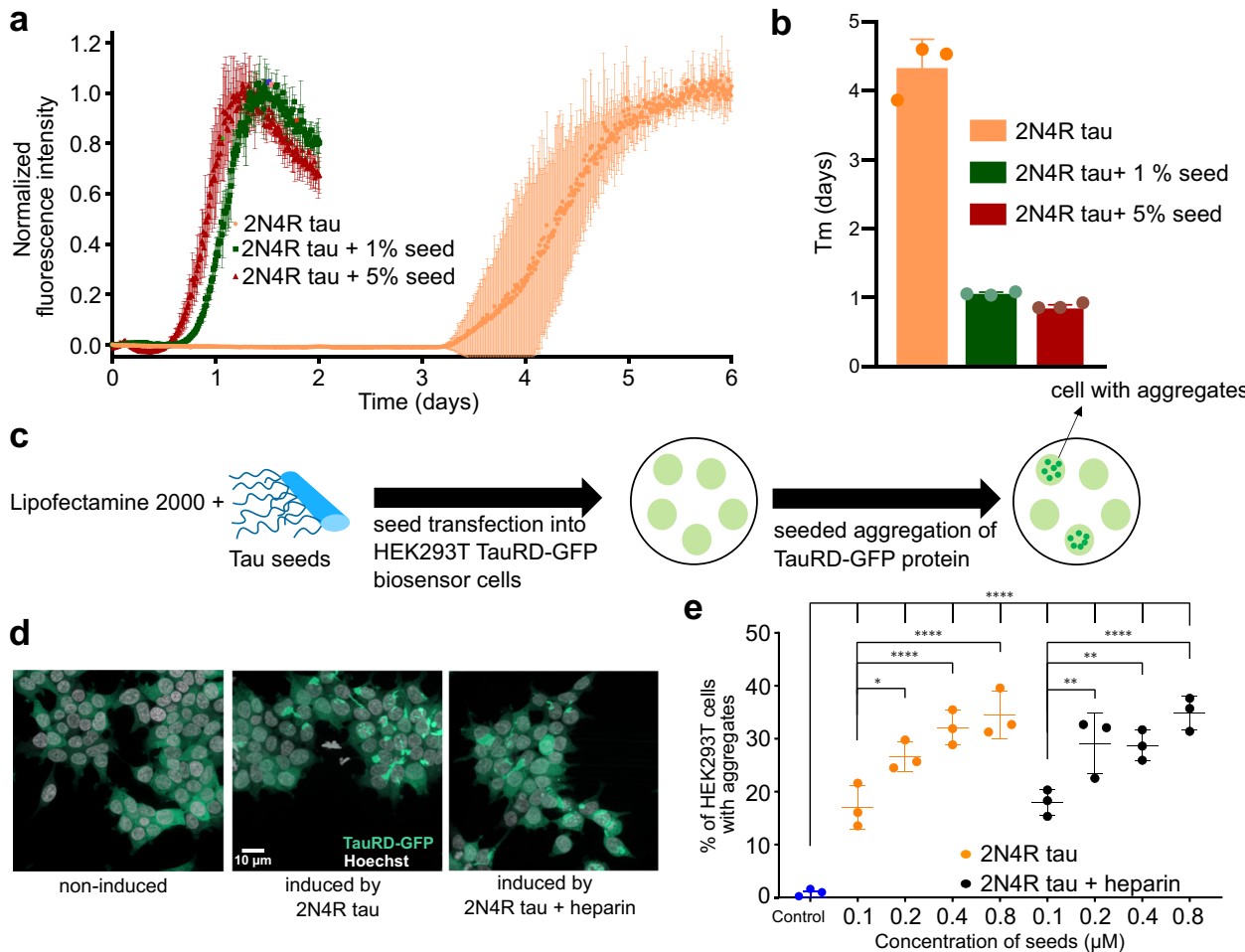

**Fig. 5 Seeding activity of heparin-free tau fibrils. a** Aggregation kinetics of 25 μM 2N4R tau in the absence (orange) and presence of 1% (green) and 5% (red) tau seeds generated without heparin. Data are presented as mean ± standard deviation of $n = 3$ independent samples. **b** Half-time (Tm) of aggregation of 2N4R tau in the absence (orange) and presence of 1% (green), 5% (red) tau seeds (see (**a**)). Data are presented as mean ± standard deviation of $n = 3$ independent samples. **c** Schematic representation of the process of seeding of tau aggregation in biosensor cells. **d** TauRD-GFP puncta in HEK293T biosensor cells[23], expressing the tau repeat domain carrying the mutations P301L and V337M, induced by 2N4R tau seeds, which were formed either in the absence (left) or presence of heparin (right). The result was reproducible for three independently performed experiments. **e** Comparison of the efficiency of 2N4R tau seeding in tau biosensor cells. Fibrils were generated by aggregating 2N4R tau in the absence (orange) or presence (black) of heparin. Different concentrations of seeds (0.1–0.8 μM) were used to induce TauRD-GFP puncta. The statistical analysis between the % of HEK293T cells with puncta induced by each concentration of seeds in the absence/presence of heparin was performed by one-way ANOVA analysis. Fibrils of different concentrations were independently transfected $n = 3$ times. Error bars represent the standard deviation of three independent experiments. Four stars represent $p < 0.0001$, two stars represents $p \leq 0.0021$, one star represents $p = 0.0296$.

contacts V350, and F378 contacts V275 (Fig. 4e). The two phenylalanine-valine cross peaks observed in the spectra of 2N4R fibrils might thus arise from these structure-specific contacts. In addition, there is only a single close contact between the aromatic ring of a histidine (H330) and the $C_\gamma$ of valine (V318) in the structure of CBD fibrils (Fig. 4e). This could correspond to the observed His($^{15}$N-ring)/V($^{13}C_\gamma$) cross peak in 2N4R fibrils (Fig. 4d).

**Seeding of tau fibrillization in vitro and in cells.** Tau pathology in the brain follows a "prion-like" behavior with transneuronal propagation of tau aggregates from one brain region to another[7]. A critical process in tau spreading is the ability of tau aggregates to seed fibrillization of monomeric tau. In order to study the in vitro seeding efficiency of 2N4R fibrils, 1% and 5% seeds (w/w) of 2N4R fibrils were added to the monomeric protein. The addition of seeds decreased the half-time of aggregation four times (Fig. 5a, b), confirming the seeding activity of 2N4R fibrils.

Next, we performed an in-cell seeding experiment using tau biosensor cells (Fig. 5c)[22,23] Four different concentrations (0.1, 0.2, 0.4, and 0.8 μM) of seeds of 2N4R fibrils, as well as heparin-induced fibrils, were used to induce aggregation. The 2N4R fibril seeds efficiently induced TauRD-GFP puncta in a concentration-dependent manner (Fig. 5d, e).

**Heparin-free tau fibrils strongly bind RNA.** Tau aggregates in human patient brains are extensively decorated with RNA[24], consistent with the positively charged surface seen in the cryoEM structure of CBD fibrils (Fig. 6a). To investigate the interaction of 2N4R fibrils with RNA, we added polyU, polyA or tRNA to the fibrils and incubated the samples for 1.5 h in the aggregation assay buffer (10 mM KCl, 25 mM HEPES, 5 mM $MgCl_2$, 3 mM TCEP, pH 7.2) followed by centrifugation to separate the pellet and supernatant (Fig. 6b). In these conditions, 2N4R tau fibrils were found to interact strongly with all three RNAs as detected by the presence of ~82% polyU, ~80% polyA and ~93% tRNA in the

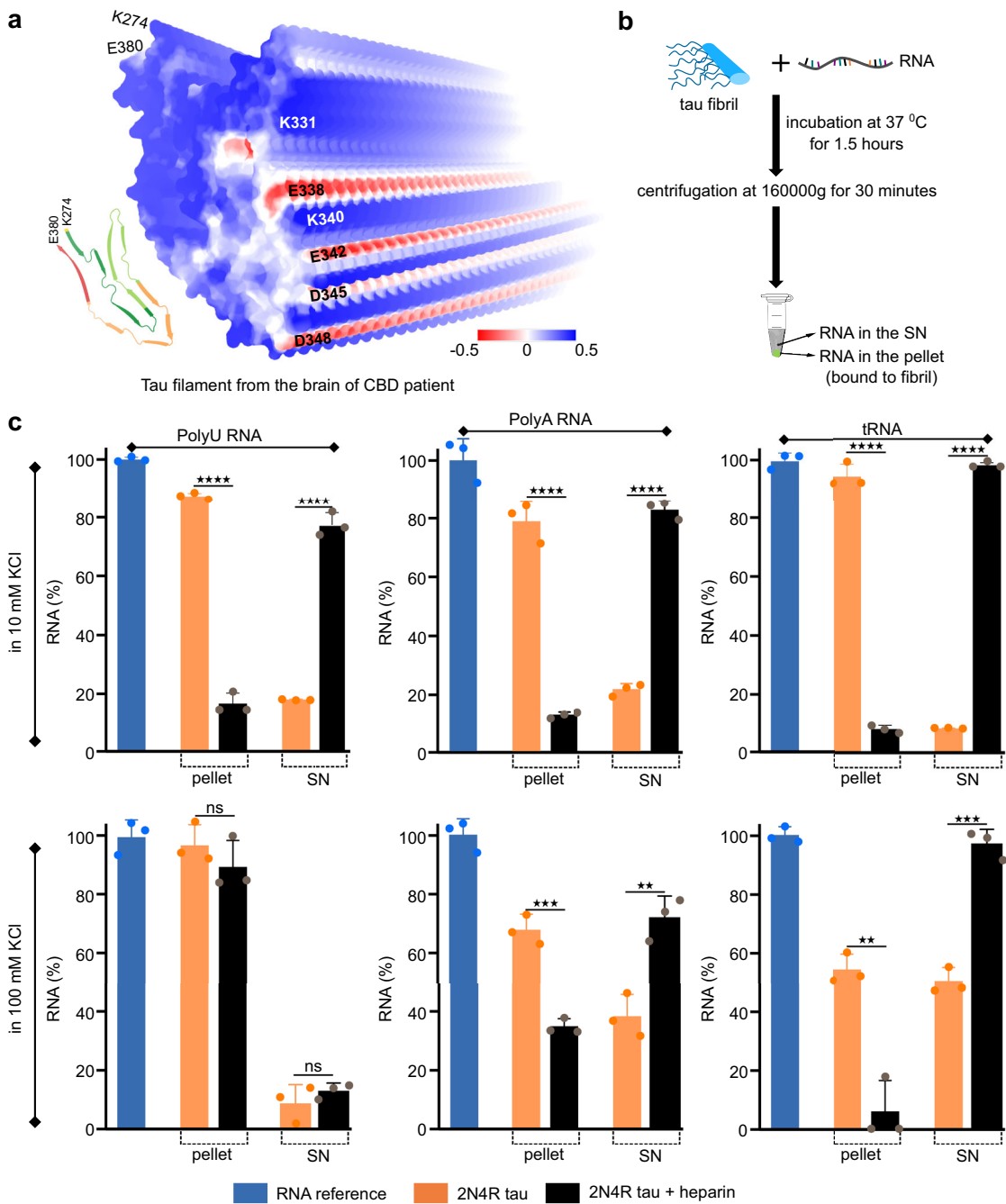

**Fig. 6 Heparin-free tau fibrils strongly bind RNA. a** Electrostatic surface potential of the tau filament extracted from a CBD patient (PDB code: 6TJO). Positively and negatively charged residues are shown in blue and red, respectively. **b** Schematic representation of the protocol to determine the binding of RNA to tau fibrils. **c** Quantitative analysis of the binding of polyU, polyA and tRNA to tau fibrils aggregated without (orange) or with heparin (black) in the aggregation assay buffer (10 mM KCl, 25 mM HEPES, 5 mM MgCl$_2$, 3 mM TCEP, pH 7.2) or at increased ionic strength (100 mM KCl, 25 mM HEPES, 5 mM MgCl$_2$, 3 mM TCEP, pH 7.2). The amount of RNA in the pellet represents the % of RNA bound to the fibril and the amount of RNA in the supernatant (SN) represents the % of unbound RNA. The statistical analysis between the % of RNA bound to heparin-free and heparin-induced tau fibrils was performed by Welch's $t$ test. Four stars represent $p < 0.0001$, three stars represents $p < 0.0002$, two stars represents $p < 0.0021$. Data are represented as mean values of $n = 3$ independent experiments. Error bars represent the standard deviation of three independent experiments.

pellet (Fig. 6c). In contrast, only ~18% polyU, ~15% polyA and ~7% tRNA were found in the pellets of the heparin-induced fibrils (Fig. 6c) suggesting very weak affinity of all three RNAs to the heparin-induced fibrils in the aggregation assay buffer.

In order to investigate the fibril interaction of the three RNAs at higher ionic strength, we repeated the experiments in a buffer with increased KCl concentration (100 mM KCl). At this higher ionic strength, the 2N4R tau fibrils, which were aggregated in

the absence of heparin, interacted with all three RNAs (Fig. 6c). Comparison of the amount of RNA in the pellets further suggested that polyU binds most efficiently to the fibrils (Fig. 6c). In the case of the heparin-induced 2N4R tau fibrils, ~90% polyU, ~35% polyA and ~6% tRNA was detected in the pellet. The heparin-induced fibrils thus bind to polyU in 100 mM KCl, but hardly interact in this condition with tRNA (Fig. 6c).

## Discussion

Increasing evidence indicates that the structure of tau fibrils differs between distinct neurodegenerative diseases. This suggests that molecular pathology based on tau fibril structure might not only help to stratify patient groups but also to develop personalized medicines to treat these devastating diseases. At the same time, we currently do not know why the structure of tau fibrils is different in different diseases and which molecular factors drive tau into disease-specific amyloid fibril structures. Insight into these questions, however, are likely to be critical for the design and development of therapeutics that work in different tauopathies.

A major bottleneck towards this goal was the high solubility of tau. Thus, a major step forward was when it was shown that tau can efficiently be fibrillized in the presence of negatively charged co-factors[8,9]. A particular effective co-factor is heparin, which was therefore widely used to study the molecular mechanisms of tau aggregation. The aggregation of tau can further be accelerated when shorter constructs comprising the repeat region are used, potentially combined with genetic mutations[25–27]. Comparison of the cryo-EM structure of heparin-induced tau fibrils and tau fibrils purified from patient brain, however, showed that the structure of heparin-induced tau fibrils differs strongly from those derived from patients[28]. This questioned the results of a huge amount of studies that used heparin-induced aggregation in order to gain insight into the molecular factors that drive tau aggregation. In addition, we are still left with the situation that we know about the presence of disease-specific tau strains, but do not know what causes the formation of these strains.

In the current study, we made an important advance towards a better understanding of the molecular factors that drive tau aggregation: we showed that the full-length tau protein can be aggregated into amyloid fibrils in the absence of heparin or other co-factors (Fig. 1). We further showed that the co-factor-free tau fibrils differ drastically in their structure and molecular properties from heparin-induced fibrils (Figs. 2–6). Solid-state NMR spectroscopy in combination with biochemical experiments revealed that the rigid/semi-rigid core of the heparin-free tau fibrils is formed by the tau residues ~270 to ~380, i.e., a similar size and location of the fibril core region as in tau fibrils purified from a CBD patient brain (Figs. 2, 3). However, the similarity in the size and location of the fibril core region does not guarantee that the heparin-free 2N4R tau fibrils prepared in vitro and the tau fibrils purified from CBD patient brain have an identical structure. In vivo tau exists in an environment rich in co-factors and also undergoes post-translational modifications including phosphorylation, ubiquitination and acetylation[29,30]. The structure of the in vitro aggregated 2N4R tau fibrils therefore might differ in the presence of co-factors (e.g., RNA, polyphosphate) in the aggregation reaction or when post-translationally modified tau is used. Consistent with this hypothesis, the fibrillar cores of tau filaments from AD and CBD display distinct patterns of acetylation, phosphorylation, trimethylation and ubiquitination[5]. Using the protocol of in vitro self-aggregation of tau, the effect of different post-translational modifications and co-factors can now be studied. In contrast, heparin is likely to largely override the impact of PTMs and other co-factors (especially when these co-factors are used at low concentrations) due to its very high negative charge.

Our study further provides single-residue evidence for the contribution of the N-terminal region of tau to pathological aggregation. The broadening of the NMR signals of the N-terminal 30 residues indicates that the corresponding residues lose their mobility during aggregation (Fig. 2). Taking further into account that conformation-specific antibodies such as Alz50 and MC1 specifically detect pathological tau in brain tissue, but require two discontinuous epitopes, one located in the repeat region (residues 313–322) and the other at the N-terminus (residues 1–18)[18], suggests that the N-terminal residues of tau bind to the cross-ß-structure of the fibrils and thereby generates the pathology-specific recognition motif. Currently, we do not know if part of the N-terminal epitope folds into ß-structure as part of this process. The interaction, however, might be more transient when compared to the interactions that stabilize the fibrillar core, because the N-terminal region of tau was so far not detected in cryo-EM studies of tau fibrils (despite its importance for antibody binding).

A potential mechanism for the connection between tau fibril structure and specific diseases could be related to differences in the interactions that each tau fibril structure has with proteins and nucleic acids. In this study, we probed the interaction of the heparin-free 2N4R fibrils with three different RNAs (polyU, polyA, tRNA) (Fig. 6). PolyU is mainly disordered and it's able to engage in both hydrophobic (through the nitrogen bases) and electrostatic interactions (through the negatively charged phosphate backbone). PolyA is known to have partial secondary structure in the presence of $Mg^{2+}$ [31]. PolyA can thus engage in some hydrophobic interactions but less efficiently than polyU. tRNA, on the other hand, has a distinct secondary structure and is unable to engage in hydrophobic interactions through its bases, because the bases are hidden inside the structure. tRNA thus mainly interacts electrostatically through its negatively charged phosphate backbone.

We observed that at low ionic strength, the heparin-free 2N4R fibrils strongly interact with all three RNAs (polyU, polyA and tRNA) (Fig. 6c). At increased ionic strength, the interaction gradually weakens from polyU to polyA and to tRNA (Fig. 6c). In contrast the heparin-induced 2N4R tau fibrils were found to interact very weakly with the three RNAs at low ionic strength, but strongly bound to polyU at higher ionic strength (Fig. 6c). In addition, a weaker interaction of the heparin-induced 2N4R tau fibrils was observed with polyA, and very little interaction with tRNA at higher ionic strength (Fig. 6c). These results can be attributed to differences in the surface charge of the two fibril types and differences in the dominating interactions at low and high ionic strength: at low ionic strength electrostatic interactions dominate, while at higher ionic strength hydrophobic interactions come into play.

The surface of the heparin-free 2N4R fibril is mostly positively charged (shown in blue, Fig. 6a) with some patches of uncharged hydrophobic residues (shown in white, Fig. 6a) giving it the possibility to interact both electrostatically and hydrophobically with different RNAs. At low ionic strength, all three RNAs (polyU, polyA and tRNA) electrostatically interact with the heparin-free 2N4R tau fibrils resulting in comparable amounts of bound RNA (Fig. 6c). At higher ionic strength, hydrophobic interactions play a bigger role and the amount of bound RNA gradually decreases from polyU to polyA and to tRNA (hydrophobic interaction strength of polyU > polyA > tRNA). In contrast, heparin-induced tau fibrils (Supplementary Fig. 7) show weak affinity for RNA at low ionic strength where electrostatic repulsion between negatively charged RNA and the negatively charged heparin is strong. At higher ionic strength, this repulsion is attenuated and hydrophobic interactions between polyU and the surface of the fibrils result in binding of polyU, but not tRNA, to the heparin-induced fibrils. The combined data suggest that the binding of RNAs depends both on the structure of the RNA and the surface electrostatic properties of different tau conformers.

Taken together our study provides an important step to reveal the connection between tau fibril structure and neuronal toxicity. In combination with high-resolution structure determination by cryo-electron microscopy and solid-state NMR spectroscopy, it sets the basis for future work to investigate the impact of post-translational modifications, one of the potential factors

determining different tau strains[5], on the three-dimensional structure of tau amyloid fibrils. Our study thus can help to elucidate the still enigmatic molecular causes that guide aggregation towards disease-specific tau strains.

## Methods

**Protein preparation.** Unlabeled and $^{13}$C/$^{15}$N-labeled 2N4R tau (hTau40, Uniprot ID 10636-8, 441 residues) were expressed in *Escherichia coli* strain BL21(DE3) from a pNG2 vector (a derivative of pET-3a, Merck-Novagen, Darmstadt) in presence of ampicilin. In case of unlabeled protein, cells were grown in 1-10 L LB and induced with 0.5 mM IPTG at $OD_{600}$ of 0.8–0.9. To obtain $^{13}$C/$^{15}$N-labeled protein, cells were grown in LB until an $OD_{600}$ of 0.6–0.8 was reached, then centrifuged at low speed, washed with M9 salts ($Na_2HPO_4$, $KH_2PO_4$ and NaCl) and resuspended in minimal medium M9 supplemented with 1 g/L $^{15}NH_4Cl$ as the only nitrogen source, 4 g/L $^{13}$C glucose as carbon source, and induced with 0.5 mM IPTG.

To obtain specifically ($^{13}C_\gamma$ valine, $^{13}$C-ring phenylalanine, $^{15}$N histidine) labeled 2N4R tau, cells were grown in LB until an $OD_{600}$ of 0.6–0.8 was reached, then centrifuged at low speed, washed with M9 salts ($Na_2HPO_4$, $KH_2PO_4$ and NaCl) and resuspended in M9 minimal medium supplemented with 0.125 g/L of L-phenylalanine (ring-$^{13}C_6$, 99%) (CLM-1055, Cambridge Isotope Laboratories), 0.15 g/L of L- valine (dimethyl-$^{13}C_2$, 99%) (CLM-9217-PK, Cambridge Isotope Laboratories), 0.125 g/L of L-histidine ($^{15}N_3$, 98%) (NLM-1513 Cambridge Isotope Laboratories). To minimize scrambling, all other amino acid types were added in unlabeled form: 0.50 g/L alanine, 0.40 g/L arginine, 0.40 g/L aspartic acid, 0.05 g/L cystine, 0.40 g/L glutamine, 0.65 g/L glutamic acid, 0.55 g/L glycine, 0.23 g/L isoleucine, 0.23 g/L leucine, 0.42 g/L lysine hydrochloride, 0.25 g/L methionine, 0.10 g/L proline, 2.10 g/L serine, 0.23 g/L threonine and 0.17 g/L tyrosine, as well as 0.50 g/L adenine, 0.65 g/L guanosine, 0.20 g/L thymine, 0.50 g/L uracil and 0.20 g/L cytosine[32].

After induction with 0.5 mM IPTG, the bacterial cells were harvested by centrifugation and the cell pellets were resuspended in lysis buffer (20 mM MES pH 6.8, 1 mM EGTA, 2 mM DTT) complemented with protease inhibitor mixture, 0.2 mM $MgCl_2$, lysozyme and DNAse I. Subsequently, cells were disrupted with a French pressure cell press (in ice cold conditions to avoid protein degradation). In the next step, NaCl was added to a final concentration of 500 mM and lysates were boiled for 20 min. Denatured proteins were removed by ultracentrifugation with 127,000 g at 4 °C for 30 min. The supernatant was dialyzed overnight at 4 °C against dialysis buffer A (20 mM MES pH 6.8, 1 mM EDTA, 2 mM DTT, 0.1 mM PMSF, 50 mM NaCl) to remove salt. The following day, the sample was filtered and applied onto an equilibrated ion exchange chromatography column and the weakly bound proteins were washed out with buffer A. Tau protein was eluted with a linear gradient of 60% final concentration of buffer B (20 mM MES pH 6.8, 1 M NaCl, 1 mM EDTA, 2 mM DTT, 0.1 mM PMSF). Protein samples were concentrated by ultrafiltration (5 kDa Vivaspin, Sartorius) and purified by reverse phase chromatography using a preparative C4 column (Vydac 214 TP, 5 µm, 8 × 250 mm). The purified protein was lyophilized and re-dissolved in the aggregation assay buffer.

**Aggregation assays.** Aggregation of 25 µM 2N4R tau was performed in 25 mM HEPES, 10 mM KCl, 5 mM $MgCl_2$, 3 mM TCEP, 0.01% $NaN_3$, pH 7.2 buffer (aggregation assay buffer). One tablet of protease inhibitor (complete, EDTA-free, Sigma Aldrich) was added to 100 mL aggregation assay buffer. The buffer was filtered through a 0.2 µm filter to remove bacterial contamination. Thioflavin T (ThT) was added to the protein at a final concentration of 50 µM to monitor aggregation kinetics. A total of 100 µL of 25 µM 2N4R tau protein with 50 µM ThT was pipetted in a well of 96 well plate (Greiner Bio-one, microplate, 96 well, PS, F-bottom, Chimney well, µClear, black, non-binding, item no – 655906) with 3 polytetrafluoroethylene beads of 2.45 mm per well. The aggregation assay was performed at 37 °C in a Tecan spark plate reader with double orbital shaking (shaking duration — 1 min, shaking amplitude — 6 mm, shaking frequency — 54 rpm) at an interval of 10 min. An excitation filter at a wavelength of 430 nm with an excitation bandwidth of 35 nm was used to excite ThT. The emission wavelength was set to 485 nm with a bandwidth of 20 nm (manual gain — 40, number of flashes — 30, integration time — 40 µs). The Z-position was calibrated using an empty well before starting each experiment. ThT fluorescence data were collected using Tecan Spark control software (v 2.2). The analysis of the aggregation data was performed using Graphpad PRISM (v 9) software.

Heparin-induced fibrillization of 25 µM 2N4R tau was achieved using the same protocol described above, but 6.25 µM heparin was added to 25 µM protein.

**Electron microscopy.** A total of 30 µL of aggregated 2N4R tau sample was centrifuged at 20,000 g for 15 min in an Eppendorf centrifuge 5424. The supernatant was removed, and the pellet was resuspended in 25 µL aggregation assay buffer. KCl was added to the redissolved aggregated sample to a final concentration of 500 mM and a final volume of 30 µL to neutralize the surface charges of the fibril. The heparin-induced fibril sample was directly imaged by transmission electron microscopy (TEM) after the aggregation. Aggregated samples were adsorbed onto carbon-coated copper grids and stained by 1% uranyl acetate solution and imaged

by CM 120 transmission electron microscope (FEI, Eindhoven, The Netherlands). Pictures were taken with a Tietz F416 CMOS camera (TVIPS, Gauting, Germany).

**Circular dichroism.** Ten µL of 25 µM 2N4R fibril were centrifuged at 20,000 g for 15 min in an Eppendorf centrifuge 5424. The supernatant was removed, and the pellet was resuspended in 50 µL of distilled water. The 2N4R tau monomer was also diluted to a final concentration of 5 µM in 50 µL of distilled water. Both the fibril and the monomer samples were transferred to a 0.02 cm pathlength cuvette. CD data were collected from 190 to 280 nm using a Chirascan-plus qCD spectrometer (Applied Photophysics, UK) at 25 °C, 1.5 s per point in 1 nm steps. The datasets were averaged from 10 repeated measurements. Spectra were baseline-corrected and smoothed with a window size of six. CDNN software v 2 (Chirascan, Applied Photophysics, UK) was used to analyze the CD spectrum of the heparin-free 2N4R tau fibrils.

**Trypsin digestion.** Fifty µL of 0.9 mg/mL 2N4R fibril (aggregated in either the absence or presence of heparin) and 0.45 mg/mL of trypsin (T8003, Sigma-Aldrich) were incubated in the aggregation assay buffer for 30 min with 1400 rpm in an thermomixer (Eppendorf) at 37 °C. The trypsin-resistant fibril core was pelleted down by centrifugation at 160,000 g for 30 min at 4 °C using a Beckman Coulter Optima MAX-XP ultracentrifuge. The supernatant was discarded and the pellet was dissolved in 10 µL of aggregation assay buffer followed by loading onto a 15% SDS PAGE gel.

**Pronase digestion.** Fifty µL of 0.8 mg/mL 2N4R fibrils (aggregated in the absence of heparin) and 0.4 mg/mL of pronase from *Streptomyces griseus* (53702, Merck-millipore) were incubated in the aggregation assay buffer in a thermomixer (Eppendorf) for 30 min at 37 °C. The pronase-resistant fibril core was pelleted down by centrifugation at 160,000 g for 30 min at 4 °C using a Beckman Coulter Optima MAX-XP ultracentrifuge. The supernatant was discarded and the pellet was dissolved in 10 µL of aggregation assay buffer followed by loading onto a 15% SDS PAGE gel.

**In-gel digestion and extraction of peptides for mass spectrometry.** The respective bands from the SDS-PAGE gels were carefully cut and kept in an Eppendorf tube. To wash the gel pieces, 150 µL of water was added and incubated for 5 min at 26 °C with 1050 rpm shaking in a thermomixer. The gel pieces were spun down and the liquid was removed using thin tips (the same washing protocol was used in all subsequent steps with different solvents). The gel pieces were washed again with 150 µL acetonitrile. After washing, the gel pieces were dried for 5 min using a SpeedVacc vacuum centrifuge. To reduce disulfide bridges, 100 µL of 10 mM DTT was added to the gel pieces and incubated for 50 min at 56 °C followed by centrifugation and removal of liquid. The gel pieces were washed again with 150 µL of acetonitrile. To alkylate reduced cysteine residues, 100 µL of 55 mM iodoacetamide were added and incubated for 20 min at 26 °C with 1050 rpm shaking followed by centrifugation and removal of liquid. Subsequently, the gel pieces were washed with 150 µL of 100 mM $NH_4HCO_3$, and then twice with 150 µL of acetonitrile and dried for 10 min in a vacuum centrifuge. The gel pieces were rehydrated at 4 °C for 45 min by addition of small amounts (2–5 µL) of digestion buffer 1 (12.5 µg/mL trypsin, 42 mM $NH_4HCO_3$, 4 mM $CaCl_2$). The samples were checked after every 15 min and more buffer was added in case the liquid was completely absorbed by the gel pieces. Twenty µL of digestion buffer 2 (42 mM $NH_4HCO_3$, 4 mM $CaCl_2$) were added to cover the gel pieces and incubated overnight at 37 °C.

To extract the peptides, 15 µL water was added to the digest and incubated for 15 min at 37 °C with 1050 rpm shaking followed by spinning down the gel pieces. Fifty µL acetonitrile was added to the entire mixture and incubated for 15 min at 37 °C with 1050 rpm shaking. The gel pieces were spun down and the supernatant (SN1) containing the extracted peptides was collected. Thirty µL of 5% (v/v) formic acid was added to the gel pieces and incubated for 15 min at 37 °C with 1050 rpm shaking followed by spinning down. Again 50 µL acetonitrile were added to the entire mixture and incubated for 15 min at 37 °C with 1050 rpm shaking. The gel pieces were spun down and the supernatant (SN2) containing the extracted peptides was collected. Both supernatants (SN1 & SN2) containing the extracted peptides were pooled together and evaporated in the SpeedVacc vacuum centrifuge. The dried peptides were resuspended in 5% acetonitrile and 0.1% formic acid and analyzed using an Orbitrap Fusion Tribrid (Thermo Fischer Scientific) instrument.

**NMR spectroscopy.** The $^1$H-$^{15}$N HSQC spectrum of 25 µM 2N4R tau monomer in the aggregation assay buffer was acquired at 278 K on an Avance III 900 MHz spectrometer (Bruker) using a 5 mM HCN Cryoprobe. The chemical shift assignments of 2N4R tau is available[13].

$^{13}$C/$^{15}$N-labeled 2N4R tau was aggregated either in the absence or presence of heparin using the protocol described above, but without ThT. Approximately 30 mg of fibrils were packed into 3.2 mm MAS rotors by ultracentrifugation. Solid-state NMR experiments were acquired at 265 K on a Avance-III 850 MHz spectrometer (Bruker) using a 3.2 mm HCN probe, and a Avance-III 950 MHz spectrometer (Bruker) using a 3.2 mm HCN probe. MAS frequency for all

measurements was 17 kHz. Two-dimensional $^1$H-$^{15}$N INEPT experiments of both fibril samples were recorded with 32 scans per point (ns), and indirect acquisition times td1 = 48 ms, td2 = 50 ms. $^{13}$C-$^{13}$C radio frequency-drive recoupling (RFDR) experiments were recorded with ns = 64, td1 = 21.5 ms, td2 = 15 ms, Tmix = 1.882 ms. The final $^{13}$C-$^{13}$C RFDR spectra are the sum of 11 datasets. The 2D NCA/NCO experiment of the heparin-free 2N4R fibrils was recorded using a TEDOR sequence with ns = 64, td1 = 21.5 ms, td2 = 75.5 ms. 3D NCACX and NCOCX spectra of heparin-free 2N4R fibrils were recorded using a TEDOR pulse sequence with ns = 8, td1 = 21.7 ms, td2 = 4.72 ms, td3 = 5.9 ms. The final 3D NCACX and NCOCX spectra used for analysis are the sum of nine experiments each. Spectra were processed using the Topspin 3.6.2 (Bruker) and analyzed using CCPNMR 2.4.2[33].

Signals in the $^1$H-$^{15}$N INEPT spectra were assigned by transferring the resonance assignment of the 2N4R tau monomer. Intensity ratios were calculated by dividing the signal intensity observed for each residue in the $^1$H-$^{15}$N INEPT spectrum of the fibril sample by the signal intensity observed in the $^1$H-$^{15}$N HSQC spectrum of the monomeric protein. The residue with the highest intensity ratio was normalized to 1. The average line connecting residues was calculated by smoothening using a 2nd order polynomial function with a window size of four.

DNP-enhanced solid-state NMR spectra were recorded for heparin-free fibrils of 2N4R labeled with $^{13}$C$_2$ (methyl) Val, and $^{13}$C$_6$ (ring) Phe and $^{15}$N$_3$ His using a commercial Bruker DNP spectrometer system with a sweepable cryo-magnet of 14.1 ± 0.1 T (~600 ± 5 MHz $^1$H frequency) and 395 GHz gyrotron for the microwaves, operating at 100 K with the Bruker 3.2 mm LT-MAS HCN probe and an Avance III HD spectrometer. 2N4R tau sample was prepared for MAS DNP NMR as follows: the 2N4R tau fibril pellet was mixed with 10 μL of the stock 10 mM TEMTriPol-1[19,20] in DNP juice (60:30:10 by volume $^{12}$C-glycerol-$d_8$: D$_2$O: H$_2$O). The excess solution was removed by ultracentrifugation at 24,000 rpm for 1 h with a Beckman Coulter SW32-Ti rotor (k-Factor 204). After removal of the excess solution, the sample was homogenized with a non-stick stainless steele needle tool. The TEMTriPol-1 final concentration was estimated at 3 mM. The sample was packed into a 3.2 mm Bruker zirconia MAS rotor with a vespel drive cap and spun at a frequency of 12.5 kHz. The 90° hard pulses were, respectively, 2.5, 3.5 and 6 μs for $^1$H, $^{13}$C and $^{15}$N. The $^1$H-$^{13}$C and $^{13}$C-$^1$H cross-polarization (CP) conditions were met with 75 kHz for $^{13}$C and a ramp from 90 to 100 kHz for $^1$H and a duration of 700 μs ($^1$H to $^{13}$C CP) or 350 μs ($^{13}$C to $^1$H CP). The $^1$H-$^{15}$N and $^{15}$N-$^1$H CP conditions were met with 30 kHz $^{15}$N radiofrequency (RF) and 46–45 kHz linear ramp for $^1$H RF with a duration of 350 μs ($^1$H-$^{15}$N) and 370 μs ($^{15}$N-$^1$H). The interscan delay was 2.5 s. The magnetic field was calibrated to an internal standard, the $^{13}$C resonance of the silicone plug (polydimethylsiloxane) used for DNP sample packing, which was set to be at 1.6 ppm on the TMS scale[34]. With these NMR settings, 2D hCHHC at 200 μs and 400 μs H-H mixing times were acquired for 2 days, and hNHHC at 200 μs H-H mixing time for 3 days[21].

**Seeding of 2N4R tau aggregation.** For in vitro seeding experiments, 2N4R tau fibrils, which had been generated in the absence of heparin according to the protocol described above, were used as seed. One percent (w/w) and 5% (w/w) seeds were added to 25 μM monomeric 2N4R tau in the aggregation assay buffer. Prior to addition, the seeds were sonicated for 2 min at 37 °C in a water bath. The same protocol was used for aggregation, as described above. The in vitro measurements were performed with three independently prepared samples.

Biosensor cell line HEK293T TauRD-GFP were generated[23] to assess the seeding activity of fibrils in cellula. These cells were engineered to stably express the human tau repeat domain (RD; amino acids 243–375) with point mutations P301L/V337M and an carboxyterminal GFP-tag connected through an 18-amino acid flexible linker (EFCSRRYRGPGIHRSPTA) (thereafter termed TauRD-GFP[35]. HEK293T TauRD-GFP cells were seeded on 384 well plates. Next day, tau fibrils were mixed with lipofectamine 2000 and added to the cells at different concentrations. Prior to addition, the fibrils were sonicated for 6 min at 10% power. After 3 days, cells were fixed and nuclei were stained with Hoechst. Images were taken with an automated confocal microscope (CellVoyager 6000) and automated image analysis was performed to calculate total Tau-GFP cell numbers and cells containing Tau-GFP aggregates[23]. Please note that automatic image analysis of aggregates takes into account local fluorescence intensity increase, resulting in low background levels of positives due to condensed cell bodies. Cell measurements were performed with three independent transfections at different concentrations per fibril type.

**RNA binding assay.** A total of 10 μg/mL of yeast tRNA (AM7119, Thermo Fischer Scientific) poly(A) (P9403, Sigma-Aldrich) or polyU (P9528, Sigma-Aldrich) were added to 100 μL of 5 μM 2N4R fibrils either in the aggregation assay buffer (10 mM KCl, 25 mM HEPES, 5 mM MgCl$_2$, 3 mM TCEP, pH 7.2) or in the same buffer but at increased ionic strength (100 mM KCl, 25 mM HEPES, 5 mM MgCl$_2$, 3 mM TCEP, pH 7.2). The fibril/RNA-mixture was incubated at 37 °C with 550 rpm shaking for 1.5 h in a Thermomixer (Eppendorf, Thermomixer comfort). After the incubation, the fibril/RNA-mixture was centrifuged at 160,000 g at 37 °C for 30 min in an ultracentrifuge (Optima MAX-XP, Beckman Coulter). The supernatant was removed and the pellet was resuspended in 100 μL of aggregation assay buffer with 2% SDS. The concentration of RNA in the supernatant and the pellet

(resuspended) was calculated using a Nanodrop 2000 spectrophotometer (Thermo Fischer Scientific). Spectra were baseline corrected using the buffer as reference. The concentrations of polyU, polyA and tRNA were calculated from a sample of 10 μg/mL RNA in 100 μL of buffer. Measurements were performed with three independently prepared samples in each case.

**Bicinchoninic acid assay.** The concentration of 2N4R tau in the supernatant after aggregation (in the absence of heparin) was determined using the micro-bicinchoninic acid (BCA) assay kit (23235, Thermo Fischer Scientific). The microplate procedure (linear working range of 2–40 μg/mL) as described in the instruction manual provided by the manufacturer was used to perform the experiment.

**Electrostatic surface potential.** The electrostatic surface potential of the tau filament (type 1) extracted from the brain of a patient with CBD (PDB code: 6TJO) and the heparin-induced 2N4R tau fibrils (snake form; PDB code: 6QJH) was displayed using the coulombic surface coloring tool in UCSF Chimera[36] (v 1.14.0). The dielectric constant of water (ε = 78.4) was used; the distance from the surface (d) was set to 1.4 Å. The minimum value for the number of colors was set to −0.5 kcal/mol*e and the maximum value was set to 0.5 kcal/mol*e.

**Reporting summary.** Further information on experimental design is available in the Nature Research Reporting Summary linked to this paper.

## Acknowledgements
We thank the mass spectrometry facility of the Max Planck Institute for Biophysical Chemistry (MPIBPC) for mass spec data, and the EM facility of MPIBPC for electron micrographs and the chemical synthesis facility of MPIBPC for preparation of the TEMTriPol-1 radical. M.Z. was supported by the European Research Council (ERC) under the EU Horizon 2020 research and innovation program (grant agreement No. 787679).

## Author contributions
P.C. conducted aggregation assays, biochemical analysis, RNA binding studies and NMR experiments; G.R. recorded solid-state NMR spectra including DNP experiments; S.L. and A.H. performed biosensor cell experiments; R.D. recorded DNP NMR experiments; A.I.O. supported biophysical analysis; L.A. supervised solid-state NMR experiments; I.M. V. supervised biosensor cell experiments; P.C. and M.Z. designed the project and wrote the paper.

## Funding

## Competing interests
The authors declare no competing interests.

## Data availability
All relevant data are available from the corresponding author upon reasonable request. Source data are provided with this paper. All the PDB codes (6TJO, 6QJH) cited in this paper are available in the protein data bank web server.

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
