## [Peer Review File · Nature Communications]

REVIEWER COMMENTS

Reviewer #1 (Remarks to the Author):

The authors describe their finding that amyloid fibrils of the tau protein can be formed in the absence of any cofactors such as heparin as done previously. They also provide a qualitative characterization of these fibrils by means of biochemical assays, CD, mass spectrometry and NMR spectroscopy. Heparin-free tau fibrils can also seed fibril growth in HEK293T TauRD-GFP cells. Finally, it is shown that heparin-free tau fibrils strongly bind RNA.

The fact that tau fibrils can be produced in the absence of heparin *in vitro* is a nice experimental finding *per se*, but its biological relevance is disputable. Furthermore, the characterization of these fibrils as reported in the manuscript remains rather qualitative. These days, detailed structures of amyloid fibrils determined by cryo-EM are reported on a regular basis. The NMR results, which certainly were achieved with great effort by an expert NMR group, overall do not provide much insight. The heparin-free fibrils seed tau in cellulo just as fine as the heparin-containing ones, which does not provide further biological insight. The fact that heparin-free fibrils bind RNA very efficiently is also not surprising; I would assume that they do so with any given anionic polyelectrolyte considering the highly positive surface charge density of tau fibrils. For the claims made in the manuscript the authors would need to show the specificity of RNA binding in comparison to for instance DNA, poly-Glu or PSS.

The discussion of the results is rather speculative, as expressed in statements like: "it is likely that the tau fibrils aggregated *in vitro* in the absence of heparin have similar β -strands as the patient-derived fibrils"

"the N-terminal residues of tau might bind to the cross- β -structure of the fibrils"
and

"In the same way, we expect the heparin-free tau fibrils to interact in a specific manner with a large variety of neuronal protein sand thus display unique properties of cell-to-cell spreading *in vivo*"

So it remains to be shown if the proposed structural similarity of the heparin-free grown tau fibrils with fibril material obtained from patients exists, and even if it did, what could be learned from it. As the authors are well aware of, amyloid fibrils are subject to a rich polymorphism and the biological meaning of it has not been found so far.

Taken together, I find the topic of the work interesting but the current manuscript does not provide sufficient novelty and biological relevance justifying publication in Nature Communications.

Reviewer #2 (Remarks to the Author):

The manuscript describes mechanistic and structural studies of tau fibrils formed in the absence of polyanionic inducer. This is an important finding that contradicts the dogma in the field tau requires an inducer in order to aggregate, enhances our understanding of tau fibril polymorphism, and should be of broad interest. In general, the experimental work is well executed and clearly described. However, there are several issues that need to be addressed:

1. The work is explicitly framed as if co-factor-free fibrils are biologically relevant. While heparin as an inducer is clearly not physiological, tau exists *in vivo* in an environment rich in RNA and other polyanions. As shown in Ref. 20 and many others, biological tau fibrils are

closely associated with such molecules. Thus I am skeptical that co-factor-free fibrils are in any way mechanistically representative of biological aggregates, and I would suggest that these conclusions should be softened.

2. Measurements of the critical concentration – the concentration of tau remaining in solution at the end of an aggregation reaction – should be presented for the data in Fig. 1D and Fig. 4A. This is an important thermodynamic parameter reflective of the effective affinity of monomer for fibril during the elongation phase, and is known to be affected by heparin. Given that the experimental conditions (very low salt, shaking with beads, temperature) were all selected to strongly favor aggregation, and the co-factor-free reaction is still fairly slow, it would be important to know whether only a small fraction of tau assembled into fibrils.

3. The seeding activity reported in vitro and in cells is quite weak. In vitro, there is poor dose dependence, with the 1% and 5% seeded reactions occurring on almost the same timescale. Given the magnitude of the change between the unseeded and 1% reactions, one would expect the 5% reaction to have a completely abrogated lag phase. Do the authors have an explanation for this? Additionally, if the unseeded and seeded reactions have different plateaus in the (non-normalized) fluorescence intensity, that should be discussed – that would be a rather atypical and interesting result. In cells, the percentage of cells without aggregates in the absence of seeding should be added to the figure. Significance testing could then be done to examine any difference in co-factor-free vs. heparin-induced seeds across all concentrations tested (rather than only within each concentration).

4. The combination of mass spectrometry and solid-state NMR to characterize the fibrils is quite elegant. However, the description of the tryptic digestion steps could be clearer. My understanding is that fibrils were digested by trypsin and then run on SDS-PAGE. The major bands were then excised and further digested by trypsin. If so, what were the trypsin concentration and time used for the second digestion step? There is also no example data or information on specificity and sensitivity of the mass spec analysis (or even the model of the instrument used). More information ought to be provided. I also have two suggestions on this part of the work: use a more promiscuous protease (e.g. proteinase K) to improve peptide coverage, particularly in the first digestion step, which should provide more resolution in the C-terminus. Second, use MS to characterize the peptides generated in the first digestion step, which will provide more direct insight into the protease accessibility of regions outside the fibril core.

Reviewer #3 (Remarks to the Author):

The manuscript by Chakraborty et al. reported structural characterization of co-factor-free 2N4R tau fibrillar aggregates that are able to self-propagate and sequester RNA. The authors also showed that the co-factor-free tau fibrillar aggregates have distinct molecular conformations from those of heparin-induced tau fibrils. Moreover, it was claimed that the in vitro aggregates may have similar structural features to those of ex vivo tau fibrils extracted from CBD patients. It would be of great interest if the in vitro tau fibrils have similar structural characteristics to those of ex vivo tau fibrils and the ex vivo tau structure can be reproduced in vitro.

However, the authors should provide more convincing evidence for the structural similarity. The only evidence they provided is the size and location of the fibril core region, which do not guarantee that the two fibrils have similar core structures. In addition, homogeneity of the

in vitro co-factor-free 2N4R tau fibrils was not discussed. The broad NMR resonances in the solid-state NMR spectra for the in vitro tau fibrils suggest the fibrils are rather highly heterogeneous (potentially polymorphic), rendering structural comparison even more challenging.

Other comments:

1. The 2N4R tau is known to be highly soluble at pH 7.4. Authors used polytetrafluoroethylene beads in 25 mM HEPES buffer (10 mM KCl, 5 mM MgCl₂, 3 mM TCEP, 0.01% NaN₃, pH 7.2). What would be the effect of the beads on the tau aggregation? Have they tried to use higher salt concentrations such as 150 mM KCl instead of 10 mM?
2. The CD spectrum of the heparin-free 2N4R tau fibrils displays β -sheet (39 %) as well as α -helical characters. What are the relative contents for α -helical and disordered regions? Which software (algorithm) was used to estimate the relative content?
3. Mass spectrometer data used for Fig. 2c should be included in the Supplemental Information.
4. In supplementary Fig. 1, all of the NMR resonances that disappeared in the fibrillar aggregates should be assigned.

Referee #1

The authors describe their finding that amyloid fibrils of the tau protein can be formed in the absence of any cofactors such as heparin as done previously. They also provide a qualitative characterization of these fibrils by means of biochemical assays, CD, mass spectrometry and NMR spectroscopy. Heparin-free tau fibrils can also seed fibril growth in HEK293T TauRD-GFP cells. Finally, it is shown that heparin-free tau fibrils strongly bind RNA. The fact that tau fibrils can be produced in the absence of heparin in vitro is a nice experimental finding *per se*, but its biological relevance is disputable. Furthermore, the characterization of these fibrils as reported in the manuscript remains rather qualitative. These days, detailed structures of amyloid fibrils determined by cryo-EM are reported on a regulatory basis. The NMR results, which certainly were achieved with great effort by an expert NMR group, overall do not provide much insight.

Reply: We thank the referee for the careful evaluation of our manuscript and highlighting the high quality of the presented data.

The heparin-free fibrils seed tau in cellulo just as fine as the heparin-containing ones, which does not provide further biological insight. The fact that heparin-free fibrils bind RNA very efficiently is also not surprising; I would assume that they do so with any given anionic polyelectrolyte considering the highly positive surface charge density of tau fibrils. For the claims made in the manuscript the authors would need to show the specificity of RNA binding in comparison to for instance DNA, poly-Glu or PSS.

Reply: Thanks for the suggestion. For the revision of the manuscript, we performed for both fibril types (non-heparin and heparin-induced) experiments with three different RNAs (polyU, polyA, tRNA) at two different ionic strength (low ionic strength and close to physiological ionic strength). The new data are shown in Fig. 6c and demonstrate that RNA binding to tau fibrils is specific and depends on both the structure of the RNA and the surface electrostatic properties of different tau conformers. For example, the heparin-induced fibrils do not bind polyU at 10 mM KCl (as would be expected for pure electrostatic interaction; black bars in Fig. 6c upper row, left panel), but do efficiently bind to polyU at 100 mM KCl (black bars in Fig. 6c lower row, left panel). In addition, tRNA binds less strongly to non-heparin fibrils than polyU RNA at 100 mM KCl (orange bars in Fig. 6c lower row). The results from these experiments are described on page 12-13 and discussed on page 16-17.

The discussion of the results is rather speculative, as expressed in statements like: “it is likely that the tau fibrils aggregated in vitro in the absence of heparin have similar β -strands as the patient-derived fibrils” “the N-terminal residues of tau might bind to the cross- β -structure of the fibrils” and “In the same way, we expect the heparin-free tau fibrils to interact in a specific manner with a large variety of neuronal protein and thus display unique properties of cell-to-cell spreading in vivo”

Reply: As suggested we removed the two statements. In addition, we now discuss the new RNA binding data, which provide insight into the underlying specificity (page 16-17), and also describe in more detail the importance of our work for studying the impact of post-translational modifications on the structure of tau amyloid fibrils (page 15-16).

So it remains to be shown if the proposed structural similarity of the heparin-free grown tau fibrils with fibril material obtained from patients exists, and even if it did, what could be learned from it. As the authors are well aware of, amyloid fibrils are subject to a rich polymorphism and the biological meaning of it has not been found so far. Taken together, I find the topic of the work interesting but the current manuscript does not provide sufficient novelty and biological relevance justifying publication in Nature Communications.

Reply: Please note that it was not the aim of our study to demonstrate that the 3D structure of the fibrils generated in the absence of heparin is identical to fibril material from patients. Instead, we show that

- full-length tau can efficiently be fibrillized in the absence of co-factors
- the structure of the non-heparin tau fibrils differs from that of heparin-induced fibrils
- the core of non-heparin fibrils is similar in size and location to that of fibril material from patients
- the N-terminal epitope of antibodies detecting pathological tau in brain tissue is immobilized in non-heparin fibrils
- non-heparin tau fibrils seed tau aggregation in biosensor cells
- RNA binding to tau fibrils is specific and depends on both the structure of the RNA and the surface electrostatic properties of different tau conformers.

Indeed the current hypothesis is that post-translational modifications (PTMs; acetylation, ubiquitination and phosphorylation) play a major role in generating different tau fibril conformations/strains (Li and Liu, *Nat Chem Biol* 2021). However, it was so far basically impossible to investigate this hypothesis, because the highly charged

heparin is likely to override the influence of PTMs in terms of aggregation kinetics and tau fibril structure. We are therefore convinced that our study is not only very novel, but is a critical step to define the molecular factors that underlie distinct tau fibril conformations in different diseases.

For the revised manuscript, we also acquired additional solid-state NMR data of the heparin-free 2N4R fibrils (shown below and in Fig. 4 of the manuscript; pages 9-11). Specifically, we recorded DNP enhanced 2D hNHHC and hCHHC spectra with specifically labeled ($^{13}\text{C}_\gamma$ valine, ^{13}C -ring phenylalanine, ^{15}N histidine) 2N4R tau fibrils. We observed contacts between the ^{13}C atoms of specific valine and phenylalanine residues, as well as between ^{13}C valine and ^{15}N histidine. These specific contacts provide further support for a structural similarity of the heparin-free *in vitro* fibrils of 2N4R tau to the CBD patient-derived tau fibrils.

Fig. 4 | Residue type-specific contacts within the core of heparin-free 2N4R tau fibrils. **a**, Proton-detected ^1H - ^{15}N correlation solid-state NMR spectrum of heparin-free fibrils of 2N4R tau selectively labeled with ^{15}N Histidine (and at the $^{13}\text{C}_\gamma$ of valine and the ^{13}C -ring of phenylalanine). The spectrum was recorded at room temperature at a NMR spectrometer with 850 MHz ^1H frequency using 55 kHz magic angle spinning (MAS) frequency. **b**, Comparison of the ^{13}C NMR signal of phenylalanine (ring carbons) when the microwave is turned on (orange) or off (black). The signal enhancement (ϵ) is displayed. Spectra shown in (b-d) were recorded at a temperature of 100 K at a NMR spectrometer with 600 MHz ^1H frequency using 12.5 kHz MAS spinning. **c**, Superposition of DNP-enhanced 2D hCHHC spectra of selectively labeled ($^{13}\text{C}_\gamma$ valine, ^{13}C -ring phenylalanine, ^{15}N histidine) heparin-free 2N4R tau fibrils with 200 μs (yellow) and 400 μs (sky blue) mixing time. Spinning side bands are labeled with *. **d**, DNP-enhanced 2D hNHHC spectrum of selectively labeled ($^{13}\text{C}_\gamma$ valine, ^{13}C -ring phenylalanine, ^{15}N histidine) heparin-free 2N4R tau fibrils. **e**, Location of valine, phenylalanine and histidine residues within the core structure of the tau filament (type 1) extracted from the brain of a patient with CBD (PDB code: 6TJO). Short-range distances between valine and phenylalanine, and histidine and valine are marked.

Referee #2

The manuscript describes mechanistic and structural studies of tau fibrils formed in the absence of polyanionic inducer. This is an important finding that contradicts the dogma in the field tau requires an inducer in order to aggregate, enhances our understanding of tau fibril polymorphism, and should be of broad interest. In general, the experimental work is well executed and clearly described.

Reply: We thank the referee for the careful evaluation of our manuscript and highlighting the importance of our study.

1. The work is explicitly framed as if co-factor-free fibrils are biologically relevant. While heparin as an inducer is clearly not physiological, tau exists *in vivo* in an environment rich in RNA and other polyanions. As shown in Ref. 20 and many others, biological tau fibrils are closely associated with such molecules. Thus I am skeptical that co-factor-free fibrils are in any way mechanistically representative of biological aggregates, and I would suggest that these conclusions should be softened.

Reply: As suggested by the reviewer we revised the discussion section. Please note, however, that the biological relevance of different factors for tau aggregation is currently unknown. As the reviewer points out, tau aggregates *in vivo* are decorated with RNA. However, similar to post-translational modifications, which could occur either prior to aggregation or once fibrils are formed, RNA and other polyanions might be less relevant for the aggregation process itself, but instead might bind to tau fibrils once these were formed. While we currently cannot distinguish between these two scenarios, we find that the heparin-free tau fibrils strongly bind to RNA. In addition, we added new data demonstrating that binding of RNA to tau fibrils is specific (Fig. 6c, pages 12-13, 16-17).

2. Measurements of the critical concentration – the concentration of tau remaining in solution at the end of an aggregation reaction – should be presented for the data in Fig. 1D and Fig. 4A. This is an important thermodynamic parameter reflective of the effective affinity of monomer for fibril during the elongation phase, and is known to be affected by heparin. Given that the experimental conditions (very low salt, shaking with beads, temperature) were all selected to strongly favor aggregation, and the co-factor-free reaction is still fairly slow, it would be important to know whether only a small fraction of tau assembled into fibrils.

Reply: Thanks for the suggestion. Quantification of the concentration of tau remaining in solution demonstrates that ~80% of tau assembled into fibrils (new Supp Fig. 1). The efficiency of fibrillization in the absence of heparin is thus comparable to that in the presence of heparin.

3. The seeding activity reported *in vitro* and in cells is quite weak. *In vitro*, there is poor dose dependence, with the 1% and 5% seeded reactions occurring on almost the same timescale. Given the magnitude of the change between the unseeded and 1% reactions, one would expect the 5% reaction to have a completely abrogated lag phase. Do the authors have an explanation for this? Additionally, if the unseeded and seeded reactions have different plateaus in the (non-normalized) fluorescence intensity, that should be discussed – that would be a rather atypical and interesting result. In cells, the percentage of cells without aggregates in the absence of seeding should be added to the figure. Significance testing could then be done to examine any difference in co-factor-free vs. heparin-induced seeds across all concentrations tested (rather than only within each concentration).

Reply: The seeding activity of both the heparin- and the non-heparin tau fibrils reached up to 40 % of total cells, which is in line with other studies using comparable cell models¹⁻⁵. We attribute the small difference between 1% and 5% seed kinetics to the strong clumping of non-heparin tau fibrils, i.e. the amount of active/available tau seeds might differ from the added volume fractions. The data for the cells in the absence of seeding have been added to Fig. 5e (including significant testing).

1. Guo, J.L. & Lee, V.M. Seeding of normal Tau by pathological Tau conformers drives pathogenesis of Alzheimer-like tangles. *J Biol Chem* **286**, 15317-15331 (2011).
2. Stohr, J. *et al.* A 31-residue peptide induces aggregation of tau's microtubule-binding region in cells. *Nat Chem* **9**, 874-881 (2017).
3. Woerman, A.L. *et al.* Tau prions from Alzheimer's disease and chronic traumatic encephalopathy propagate in cultured cells. *Proc Natl Acad Sci U S A* **113**, E8187-E8196 (2016).
4. McEwan, W.A. *et al.* Cytosolic Fc receptor TRIM21 inhibits seeded tau aggregation. *Proc Natl Acad Sci U S A* **114**, 574-579 (2017).
5. Nachman, E. *et al.* Disassembly of Tau fibrils by the human Hsp70 disaggregation machinery generates small seeding-competent species. *J Biol Chem* **295**, 9676-9690 (2020).

The unseeded and seeded *in vitro* reactions have comparable plateaus in non-normalized ThT intensity:

4. The combination of mass spectrometry and solid-state NMR to characterize the fibrils is quite elegant. However, the description of the tryptic digestion steps could be clearer. My understanding is that fibrils were digested by trypsin and then run on SDS-PAGE. The major bands were then excised and further digested by trypsin. If so, what were the trypsin concentration and time used for the second digestion step? There is also no example data or information on specificity and sensitivity of the mass spec analysis (or even the model of the instrument used). More information ought to be provided. I also have two suggestions on this part of the work: use a more promiscuous protease (e.g. proteinase K) to improve peptide coverage, particularly in the first digestion step, which should provide more resolution in the C-terminus. Second, use MS to characterize the peptides generated in the first digestion step, which will provide more direct insight into the protease accessibility of regions outside the fibril core.

Reply: We have extended the description of the tryptic digestion/MS experiments in the revised version of the manuscript. In addition, we added data for pronase treatment (i.e. a mixture of proteases) (new Supp Fig. 4). Regarding the suggestion to characterize the peptides in the first digestion step by MS, this is tricky because it is difficult to detect trypsin-digested peptides from the N-terminal half of tau by MS (please see our previous study in which we extensively used MS on tau: Ukmar-Godec et al. *Sci Adv* 2020). There is also a second problem: To characterize the peptides generated in the first step of digestion, one has to make sure that the supernatant (which contains monomeric protein) is completely removed from the pellet. The presence of even a small amount of monomeric protein from the supernatant is enough to detect peptides in MS.

Referee #3

The manuscript by Chakraborty et al. reported structural characterization of co-factor-free 2N4R tau fibrillar aggregates that are able to self-propagate and sequester RNA. The authors also showed that the co-factor-free tau fibrillar aggregates have distinct molecular conformations from those of heparin-induced tau fibrils.

Reply: We thank the referee for the careful evaluation of our manuscript.

Moreover, it was claimed that the in vitro aggregates may have similar structural features to those of ex vivo tau fibrils extracted from CBD patients. It would be of great interest if the in vitro tau fibrils have similar structural characteristics to those of ex vivo tau fibrils and the ex vivo tau structure can be reproduced in vitro. However, the authors should provide more convincing evidence for the structural similarity. The only evidence they provided is the size and location of the fibril core region, which do not guarantee that the two fibrils have similar core structures.

Reply: To obtain additional information about the structure of the 2N4R fibrils, we recorded DNP-enhanced 2D hNHHHC and hCHHC spectra with specifically labeled ($^{13}\text{C}_\gamma$ valine, ^{13}C -ring phenylalanine, ^{15}N histidine) 2N4R tau fibrils (shown below and in Fig. 4 of the manuscript; pages 9-11). We observed contacts between the ^{13}C atoms of specific valine and phenylalanine residues, as well as between ^{13}C valine and ^{15}N histidine. These specific contacts suggest some local structural similarity of the heparin-free in vitro fibrils of 2N4R tau to the CBD tau fibrils.

Fig. 4 | Residue type-specific contacts within the core of heparin-free 2N4R tau fibrils. **a**, Proton-detected ^1H - ^{15}N correlation solid-state NMR spectrum of heparin-free fibrils of 2N4R tau selectively labeled with ^{15}N Histidine (and at the $^{13}\text{C}_\gamma$ of valine and the ^{13}C -ring of phenylalanine). The spectrum was recorded at room temperature at a NMR spectrometer with 850 MHz ^1H frequency using 55 kHz magic angle spinning (MAS) frequency. **b**, Comparison of the ^{13}C NMR signal of phenylalanine (ring carbons) when the microwave is turned on (orange) or off (black). The signal enhancement (ϵ) is displayed. Spectra shown in (b-d) were recorded at a temperature of 100 K at a NMR spectrometer with 600 MHz ^1H frequency using 12.5 kHz MAS spinning. **c**, Superposition of DNP-enhanced 2D hCHHC spectra of selectively labeled ($^{13}\text{C}_\gamma$ valine, ^{13}C -ring phenylalanine, ^{15}N histidine) heparin-free 2N4R tau fibrils with 200 μs (yellow) and 400 μs (sky blue) mixing time. Spinning side bands are labeled with *. **d**, DNP-enhanced 2D hNHHC spectrum of selectively labeled ($^{13}\text{C}_\gamma$ valine, ^{13}C -ring phenylalanine, ^{15}N histidine) heparin-free 2N4R tau fibrils. **e**, Location of valine, phenylalanine and histidine residues within the core structure of the tau filament (type 1) extracted from the brain of a patient with CBD (PDB code: 6TJO). Short-range distances between valine and phenylalanine, and histidine and valine are marked.

Please also note that it is not the primary aim of our study to demonstrate that the 3D structure of the fibrils generated in the absence of heparin is identical to fibril material from patients. Instead, we show that

- full-length tau can efficiently be fibrillized in the absence of co-factors
- the structure of the non-heparin tau fibrils differs from that of heparin-induced fibrils
- the core of non-heparin fibrils is similar in size and location to that of fibril material from CBD patients
- the N-terminal epitope of antibodies detecting pathological tau in brain tissue is immobilized in non-heparin fibrils
- non-heparin tau fibrils seed tau aggregation in biosensor cells
- RNA binding to tau fibrils is specific and depends on both the structure of the RNA and the surface electrostatic properties of different tau conformers.

Indeed the current hypothesis is that post-translational modifications (PTMs; acetylation, ubiquitination and phosphorylation) play a major role in generating different tau fibril conformations/strains (Li and Liu, *Nat Chem Biol* 2021). However, it was so far basically impossible to investigate this hypothesis, because the highly charged heparin is likely to override the influence of PTMs in terms of aggregation kinetics and tau fibril structure. We are therefore convinced that our study is a critical step to define the molecular factors that underlie distinct tau fibril conformations in different diseases.

In order to highlight these aspects, we have added new data (e.g. RNA binding specificity shown in Fig. 6) and revised the discussion section. In particular, we stress the potential importance of PTMs for the formation of different tau fibril conformations (page 15-16).

In addition, homogeneity of the in vitro co-factor-free 2N4R tau fibrils was not discussed. The broad NMR resonances in the solid-state NMR spectra for the in vitro tau fibrils suggest the fibrils are rather highly heterogeneous (potentially polymorphic), rendering structural comparison even more challenging.

Reply: The isolated peaks in the 2D RFDR spectra of heparin-free 2N4R fibrils show ^{13}C line widths between 0.6 to 0.8 ppm and in 2D NCA spectra ^{15}N line widths between 0.9 to 1.1 ppm. These line widths are comparable to the previously reported line widths for monomorphic heparin-induced 0N4R tau fibrils, where ^{13}C and ^{15}N line widths of 0.5 to 0.7 ppm and 1.1 to 1.5 ppm, respectively, were reported (Dregni et al., *PNAS* 2019). This suggests that the 2N4R tau fibrils generated in the absence of heparin are structurally homogeneous (at least not particularly polymorphic). We included the observed line widths in the updated manuscript (page 7-8).

Other comments:

1. The 2N4R tau is known to be highly soluble at pH 7.4. Authors used polytetrafluoroethylene beads in 25 mM HEPES buffer (10 mM KCl, 5 mM MgCl₂, 3 mM TCEP, 0.01% NaN₃, pH 7.2). What would be the effect of the beads on the tau aggregation? Have they tried to use higher salt concentrations such as 150 mM KCl instead of 10 mM?
2. The CD spectrum of the heparin-free 2N4R tau fibrils displays β -sheet (39 %) as well as α -helical characters. What are the relative contents for α -helical and disordered regions? Which software (algorithm) was used to estimate the relative content?
3. Mass spectrometer data used for Fig. 2c should be included in the Supplemental Information.
4. In supplementary Fig. 1, all of the NMR resonances that disappeared in the fibrillar aggregates should be assigned.

Reply: Thanks for these additional comments. We have implemented the suggested changes. The relative content of random coil is 31%, α -helix is 14% as estimated by CDNN (Chirascan, Applied photophysics) (now stated in the methods section). Regarding the influence of beads and ionic strength: no aggregation occurred without beads, increasing the KCl concentration from 10 to 100 mM slows down the aggregation (three replicates each):

REVIEWER COMMENTS

Reviewer #1 (Remarks to the Author):

The authors have in part addressed and answered my concerns.

However, I may not have made my point clear that heparin-free tau fibrils would most likely bind to any negatively charged polyelectrolyte such as DNA, polyGlu, PSS, dextran sulfate or similar. What the authors have done is that they used further RNA molecules to demonstrate that RNA binding is specific for heparin-free tau fibrils, but the proper control would be to study interaction of heparin-free tau fibrils with non-RNA-like negatively charged polyelectrolytes as named above. Without such a control, I do not find the statement that heparin-free fibrils specifically bind RNA justified.

Also, the CD spectrum of the heparin-free fibrils looks more α -helical than β -sheet like, how much α -helix content was determined from the analysis of the CD spectra (the authors only report 39% β -sheet).

Reviewer #2 (Remarks to the Author):

All of my concerns have been well addressed.

Reviewer #3 (Remarks to the Author):

Authors properly responded to the reviewer's concern and comments.

Referee #1

The authors have in part addressed and answered my concerns. However, I may not have made my point clear that heparin-free tau fibrils would most likely bind to any negatively charged polyelectrolyte such as DNA, polyGlu, PSS, dextran sulfate or similar. What the authors have done is that they used further RNA molecules to demonstrate that RNA binding is specific for heparin-free tau fibrils, but the proper control would be to study interaction of heparin-free tau fibrils with non-RNA-like negatively charged polyelectrolytes as named above. Without such a control, I do not find the statement that heparin-free fibrils specifically bind RNA justified.

Reply: We did not want to imply that the binding of RNA to the heparin-free fibrils is specific when compared to non-RNA-like negatively charged polyelectrolytes. We therefore changed/reduced the statement on page 9 to *“The combined data suggest that the binding of RNAs depends both on the structure of the RNA and the surface electrostatic properties of different tau conformers.”* This statement summarizes our findings shown in Fig. 6.

Also, the CD spectrum of the heparin-free fibrils looks more α -helical than β -sheet like, how much α -helix content was determined from the analysis of the CD spectra (the authors only report 39% β -sheet).

Reply: The α -helix content determined from the CD spectrum of the heparin-free fibrils is 14%. The appearance of the CD spectrum is influenced by the contribution from the tau regions that remain disordered in the fibrils (termed “fuzzy coat”).